# Thermo-Tunable Pores and Antibiotic Gating Properties of Bovine Skin Gelatin Gels Prepared with Poly(n-isopropylacrylamide) Network

**DOI:** 10.3390/polym12092156

**Published:** 2020-09-22

**Authors:** Fang-Chang Tsai, Chih-Feng Huang, Chi-Jung Chang, Chien-Hsing Lu, Jem-Kun Chen

**Affiliations:** 1Department of Materials and Science Engineering, National Taiwan University of Science and Technology, 43, Sec. 4, Keelung Road, Taipei 106, Taiwan; tfc0323@gmail.com; 2Department of Chemical Engineering, i-Center for Advanced Science and Technology (iCAST), National Chung Hsing University, Eng Bld 3, 145 Xingda Road, Taichung 40227, Taiwan; huangcf@nchu.edu.tw; 3Department of Chemical Engineering, Feng Chia University, 100, Wenhwa Road, Seatwen, Taichung 40724, Taiwan; changcj@fcu.edu.tw; 4Department of Obstetrics and Gynecology, Taichung Veterans General Hospital, 1650 Taiwan Boulevard Sect. 4, Taichung 40705, Taiwan; 5R&D Center for Membrane Technology, Chung Yuan Christian University, Chungli, Taoyuan 32043, Taiwan

**Keywords:** poly(*N*-isopropylacrylamide), bovine skin gelatin, porous hydrogel, polystyrene nanosphere

## Abstract

Polystyrene nanospheres (PNs) were embedded in bovine skin gelatin gels with a poly(*N*-isopropylacrylamide) (PNIPAAm) network, which were denoted as NGHHs, to generate thermoresponsive behavior. When 265 nm PNs were exploited to generate the pores, bovine skin gelatin extended to completely occupy the pores left by PNs below the lower critical solution temperature (LCST), forming a pore-less structure. Contrarily, above the LCST, the collapse of hydrogen bonding between bovine skin gelatin and PNIPAAm occurred, resulting in pores in the NGHH. The behavior of pore closing and opening below and above the LCST, respectively, indicates the excellent drug gating efficiency. Amoxicillin (AMX) was loaded into the NGHHs as smart antibiotic gating due to the pore closing and opening behavior. Accordingly, *E. coli*. and *S. aureus* were exploited to test the bacteria inhibition ratio (BIR) of the AMX-loaded NGHHs. BIRs of NGHH without pores were 48% to 46.7% at 25 and 37 °C, respectively, for *E. coli* during 12 h of incubation time. The BIRs of nanoporous NGHH could be enhanced from 61.5% to 90.4% providing a smart antibiotic gate of bovine skin gelatin gels against inflammation from infection or injury inflammation.

## 1. Introduction

Skin burns mostly generate the disruption of the epidermal barrier, combined with the denaturation of proteins and lipids, which is significantly prone to infection due to a fertile environment that is rich in bacterial nutrients for microbial growth [1]. The most abundant species to colonize burn wounds among species present in normal skin microflora is Staphylococcus aureus (6.5–37.6%) [2]. Antibacterial agents for the treatment of skin infections, including burns, in hospitals have favored the emergence of smart antibiotic releasing during serious opportunistic infection; therefore, it is essential [3].

Hydrogels are three-dimensionally cross-linked polymer networks with hydrogen bonding interaction; they have been exploited not only to enhance cell affinity but also sustain drug dosing in tissue engineering and biomedical applications [4,5,6]. Hydrogels also feature hydrophilicity, biocompatible, stability, and a high swelling ratio, which can apply in localized controlled drug delivery and cell culture scaffold [7,8]. Moreover, porous hydrogels can promote the surface contact area, provide narrow pore size distributions, and blend readily with other polymers for functionalization, which represent excellent types of Implantable drug delivery systems (IDDSs) [9,10,11]. Porous hydrogels encapsulate biomolecules and release them in specific conditions by blending stimuli-responsive polymers by varying their interaction of hydrogen bond, which can promote drug targeting in the specific condition to reduce potential side effects. Introducing stimuli-responsiveness into porous hydrogels can endow these hydrogels with the prominent advantages of controllable drug delivery, adjustable pore diameter, and tunable ion absorption and cell concentration, which are significantly attractive in various applications [12]. There are many fundamental studies and applications of stimuli-responsive porous hydrogels in fields such as tissue engineering [13], self-healing [14,15], supercapacitors [16], 3D printing [17,18], and metal ion collection [19]. However, the low reliability, expensiveness, and low stimulus-response efficiency of these porous hydrogels still limit their application in the abovementioned fields. Furthermore, the relaxation of the polymer network is affected by the humidity, gel size, and spinodal decomposition during phase separation, varying the stimulus-response efficiency [20,21]. Although incorporating additional porosity into the polymer network of hydrogels is a sophisticated technique to improve their stimulus-response efficiency, there is still no proven method to achieve a high stimulus-response efficiency of pore opening and closing [22].

The most common method to form regular porous structures in thermoresponsive polymers is dispersing polystyrene nanospheres (PNs) in an aqueous solution of these polymers. The association between PN and poly(*N*-isopropylacrylamide) (PNIPAAm) has already been used to develop various patterns for biomedical applications [23,24]. Smart gating membranes possessing regular-sized pores/channels can self-modify their permeability degree by adjusting the pore/channel size in response to external stimuli [25,26]. Grafting or blending stimuli-responsive polymers in/on the pores/channels of porous membranes is a general strategy to produce stimuli-responsive smart gating membranes [27,28]. To date, the complicated synthesis routes, the limited stimulus-response efficiency, or poor mechanical properties have limited the development of smart gating membranes for industrial applications. To achieve mass production, the development of a feasible approach for the preparation of smart gating membranes with high flux, excellent mechanical properties, and satisfied responsiveness is a next-generation goal of scientists. Additionally, blending natural components with stimuli-responsive polymers through inter-hydrogen bonding would be a possible strategy to improve the biocompatibility of membranes. Gelatin, a nature polymer, is composed of rich amino acid groups, which has good biocompatibility and no cytotoxicity [29]. Compared with other natural polymers, gelatin can form inter-hydrogen bonds of “appropriate strength” with PNIPAAm. The high strength of inter-hydrogen bonds may retard the switching from the inter- to intra-hydrogen bonding of PNIPAAm with gelatin. In our previous study, switching of the inter- and intra-hydrogen bonding between gelatin and pH-responsive hydrogels was reported to be correlated with biodegradability, biocompatibility, and controlled drug delivery [30].

Herein, we combined the upper critical solution temperature (UCST) behavior of gelatin and LCST behavior of PNIPAAm to develop a thermoresponsive PNIPAAm–gelatin hybrid hydrogels (NGHHs) for the smart gating of drug molecules. Cross-linked PNIPAAm was polymerized with bovine skin gelatin and a series of PNs including 265 ± 5, 385 ± 5, 790 ± 5, and 1080 ± 5 nm of particle size. The bovine skin gelatin was retained in the 3D network of cross-linked PNIPAAm to improve the mechanical properties of NGHHs. A series of NGHH scaffolds with thermo-tunable pores was obtained after the removal of PNs. During removal of the PNs, the gelated gelatin networks with stretched PNIPAAm chains in the NGHH membranes shrunk the pore sizes below both UCST of gelatin and LCST of PNIPAAm; meanwhile, the pore sizes were enlarged by solating gelatin networks with coiled PNIPAAm chains above both the UCST of gelatin and LCST of PNIPAAm. A series of porous NGHH membranes exhibited pore closing and opening behavior between 25 and 45 °C by combining the UCST behavior of gelatin and LCST behavior of PNIPAAm, which could be applied for biocompatible smart drug gating. Hereby, the efficiency of thermoresponsive pores has been enhanced significantly. The prepared thermoresponsive NGHH prolonged a moderate antibiotics release at 25 °C and sustained a high dosage of antibiotics under 37 °C conditions (simulating an inflammation and fever environment) with the various interaction degrees of hydrogen bonding between antibiotics and NGHH. Therefore, our thermoresponsive NGHHs are able to release antibiotics against bacteria in the local inflammation or fever environment under infection.

## 2. Experimental Section

### 2.1. Materials

Styrene and potassium persulfate (KPS) were obtained from Acros-Organics and used for emulsifier-free emulsion polymerization after purification. Gelatin (obtained from bovine skin, type B; the degree of substitution of amino groups was 96.7%), *N*,*N*′-methylene bisacrylamide (MBA), 2,2′-azobis(2-methylpropionitrile) (AIBN), and N-isopropylacrylamide (NIPAAm) were purchased from Sigma-Aldrich (St. Louis, MO, US). NIPAAm was obtained from toluene/hexane (50%, *v*/*v*) and dried in a vacuum oven before use. Toluene and other solvents (reagent grade) were purchased from Aldrich Chemical and used without further purification. Distilled water used for the experiments was prepared in our local laboratory. Amoxicillin (AMX; ≈365.404 g/mol molecular weight, molecular formula: C_16_H_19_N_3_O_5_S, purity: 96%) and methylene blue (MB; 373.90 g/mol molecular weight, molecular formula: C_16_H_24_ClN_3_O_3_S) were also purchased from Sigma-Aldrich. Lyophilized cells of Escherichia coli (*E. coli*; Strain K12) and buffered aqueous suspension of Staphylococcus aureus (*S. aureus*; Wood 46 strain) were purchased from Sigma-Aldrich.

### 2.2. Synthesis of Porous NGHH Membranes

Scheme 1 depicts the preparation of porous NGHH for smart drug releasing. PNs were synthesized according to previous studies [31]. Styrene (0.6 mol) was added to deionized water (450 mL) containing NaCl (6.75 mmol) under a nitrogen atmosphere at 70 °C over 30 min. Sequentially, potassium persulfate (KPS) (0.99 mmol) was added dropwise to this mixture under stirring at 300 rpm for emulsifier-free emulsion polymerization for 12, 24, 36, and 48 h (polymerization time), which were denoted as PN1, PN2, PN3, and PN4. The as-prepared colloidal solution was cooled to room temperature and ultracentrifuged for 20 min to obtain PNs from the solution. The PNs were redispersed in deionized water three times to remove the unreacted components. NIPAAm (1 g) and AIBN (0.014 g) were added to 1 wt % gelatin aqueous solutions (100 mL) under a nitrogen atmosphere at room temperature for 6 h to obtain NGHH solutions. PN1, PN2, PN3, and PN4 (2.5 g) and MBA (0.6 g) were added to these NGHH solutions under stirring for 12 h at room temperature.

The as-prepared samples were coated on a Teflon plate to post-bake for 24 h. Then, these NGHH scaffolds including PN1, PN2, PN3, and PN4 were immersed in toluene for 2 h to remove PNs from them, and the resulting samples were denoted as NG1, NG2, NG3, and NG4, respectively. In addition, a NGHH was cross-linked by MBA without embedding PNs, which was denoted as NG0. The properties of PNs, including particle size distribution and zeta potential, were measured by dynamic light scattering (DLS, Zetasizer, Nano ZS90). Functional groups and hydrogen bonding interactions within the samples were investigated by Fourier transform infrared spectroscopy (FTIR, Digilab, FTS-1000). These samples were separately incubated in a phosphate buffered saline (PBS) solution (pH = 7.4) at 25 (below the LCST) and 37 °C (above the LCST) for 2 h for lyophilization by the commercial freeze dryer (EYELA FDU-1200) under 1.5 mTorr at the average cooling rate of 0.8 °C/min between −10 and −50 °C. The samples lyophilized at 25 and 37 °C were sputtered with an ultrathin Pt layer and mounted on aluminum stubs to improve the image quality during morphology observation using high-resolution scanning electron microscopy (HR-SEM, JEOL JSM-6500F, Tokyo, Japan). The hydrophilicity of NGHH was evaluated by static water contact angles (SWCAs). First, 9 μL droplets of DI water were dropped on the flattened surface of the sample through a micro syringe at room temperature. The droplet images on the surface were captured to determine the contact angle using a contact angle meter (Sindatek Instruments, New Taipei City, Taiwan) on an aluminum stage where the temperature was adjusted at 25 °C (below the LCST) or 37 °C (above the LCST) using a water bath. Five tests were performed to determine the average value.

### 2.3. Characterization of the Stability and Biocompatibility of NGHHs

The degradation rate was calculated gravimetrically to evaluate the stability of the samples before and after immersion in PBS at 25 and 45 °C for different periods of time until 20 days. L929 fibroblasts (ATCC) as a model cell line were used to evaluate the in vitro biocompatibility of the NGHH membranes [32]. Initially, these as-prepared membranes were cut into 8 mm diameter discs with ≈1.5 mm thickness, washed three times with the PBS solution, immersed in 75% ethanol for sterilization, and then placed in 48-well plates. Additionally, the L929 fibroblasts were grown and maintained in a buffer solution (RPMI 1640, Gibco, 61870-010), which was refreshed every second day, at 37 °C under a 5% carbon dioxide and humidified atmospheric condition. Cells at a 9 × 10^4^ cells mL^−1^ concentration were seeded in 48-well plates and incubated to confluence on the NGHH membranes at 37 °C under the above-mentioned condition in the cell culture medium that was refreshed under the same condition after 24 h of incubation. The filtered extracts of the medium were removed at different time periods until 30 days to determine the biocompatibility of the NGHH membranes by 3-(4,5-dimethylthiazol-2-yl)-2,5-diphenyl tetrazolium bromide (MTT) assay [33]. Blank cross-linked gelatin discs were used as a negative control to calculate the relative cellular viability (%) of each sample at each incubation time. Herein, three samples were measured in triplicate to obtain the degradation rate and relative cellular viability, and the data for each sample are expressed as mean values ± standard deviations.

### 2.4. Swelling Ratio of NGHH

As-prepared NGHHs after lyophilization were exploited to evaluate gravimetrically the swelling ratios (*R_s_*) of NGHHs by immersing them in phosphate buffer solutions (PBS) at various temperatures from 25 to 51 °C after 24 h to reach the swelling equilibrium [34]. The swollen NGHHs were withdrawn from the PBS solution to remove excess solution from the samples and weighed to calculate the *R_s_* of the NGHH as follows:(1)$$R_{s}=\frac{W_{s}{-W}_{d}}{W_{d}}\times 100\%$$
where *W_d_* and *W_s_* are the weights of the samples in the dry and swelling state. Each experiment was made in triplicate, and the average was expressed as the mean ± standard deviation of the mean.

### 2.5. Drug Loading and Releasing

Lyophilized NGHH (50 mg) was immersed in drugs (10 mL) including MB and amoxicillin (AMX) for 24 h at room temperature to evaluate the ability of drug loading into the NGHHs. Drug-loaded NGHHs were withdrawn from the MB and AMX and then lyophilized. Loading density (*D*_load_) in the NGHHs is defined herein as [35]
(2)$$D_{\mathrm{load}}\frac{\mathrm{Total}\;\mathrm{drug}-\mathrm{Residual}\;\mathrm{drug}}{\mathrm{Sample}\;\mathrm{weight}}.$$

The release of MB and AMX from the NGHH scaffold was recorded at various temperatures in PBS solution (pH = 7.4) using the UV-Vis spectrophotometer. *D*_load_ of the NGHH scaffolds were obtained with the residual MB and AMX solution using a UV-Vis spectrometer (Varian-Cary 100) at 25 and 37 °C, respectively. Real-time drug release from the NGHH was tested alternatively in water bath at 25 and 37 °C: from 25 °C for 120 min, to 37 °C for 180 min, to 25 °C for 180 min, to 37 °C for 240 min. The cumulative drug release ratio is calculated by the following equation:(3)$$Cumulative\;released\;ratio\;\left(R_{cr}\right)\;\left(\%\right)=\frac{C_{t}}{C_{\infty }}\times 100\%$$
where *C*_∞_ and *C_t_* represent the final drug concentration at the end of real-time drug release and the cumulative concentration of drug release within the period *t* [36].

### 2.6. Antibacterial Activity Testing

The NGHHs were sterilized by 75% ethyl alcohol overnight and then thoroughly washed with sterilized PBS solution three times. Sequentially, the NGHHs were irradiated under UV light, and irradiation was exploited to further sterilize in a bio-safety hood for 1 h. *E. coli* and *S. aureus* were exploited as models of Gram-negative and -positive bacteria, respectively, to examine antibacterial property. As-prepared NGHHs were cut into 8 mm diameter discs with ≈1.5 mm thickness to immerse in an AMX solution for loading within the NGHHs. Then, 1 mL of bacterial suspension in PBS solution including 6 × 10^7^ cells was cultured with AMX-loaded NGHH for various periods at 25 and 37 °C, respectively, under agitation at 25 and 37 °C (100 rpm) [37]. After various incubation periods, the NGHHs were removed from the solution. The residual bacterial suspension was incubated in a 96-well culture plate under agitation at 25 and 37 °C (160 rpm) for a predetermined time interval. Each experiment was made in triplicate. In addition, the bacterial suspension was also cultured without AMX-loaded NGHH to obtain the bacterial amount as a control for blank test. The bacterial amount was determined by UV-vis spectroscopy at various conditions with average optical densities (ODs) of these bacteria at 600 nm. The bacteria inhibition ratio (*BIR*) was calculated by the following equation:(4)$$BIR\;\left(\%\right)=\frac{\mathrm{Ic}-\mathrm{Is}}{\mathrm{Ic}}\times 100\%$$
where Is and Ic represent the ODs of the bacterial suspension with and without (blank) samples for various incubation periods. Each sample was performed in triplicate to determine the average *BIR*, which is expressed as the mean ± standard deviation of the mean.

## 3. Results and Discussion

### 3.1. Characterization of the Thermoresponsive Porous NGHH

To investigate the interactions between PNIPAAm and gelatin, pure PNIPAAm, bovine skin gelatin, and NG0 were analyzed by FTIR, and their spectra are shown in Figure 1. The spectrum of PNIPAAm showed all absorption bands of PNIPAAm, i.e., 3132 and 3309 cm^−1^ (NH stretching vibrations), 2978 cm^−1^ (asymmetric stretching vibration of CH_3_), 1450 cm^−1^ (asymmetric bending vibration of CH_3_), 1670 cm^−1^ (amide-I mode), 1597 cm^−1^ (amide-II modes) [36]. In the spectrum of bovine skin gelatin, a broad band ranging from 3109 to 3695 cm^−1^ was observed, corresponding to the OH and NH stretching vibrations. The other bands at 2978, 1689, and 1238 cm^−1^ were attributed to the stretching vibrations of the CH, C=O, and C-N functional groups of bovine skin gelatin. The bands corresponding to the bending vibrations of NH and CH_2_ appeared at 1519 and 1462 cm^−1^, respectively [38]. In the spectrum of NG0, the absorption band around 3464 cm^−1^ became broader, indicating hydrogen bond formation between the –OH and –NHCO groups. Moreover, the bands at 1670 and 1597 cm^−1^ in the spectrum of PNIPAAm shifted to 1658 and 1558 cm^−1^, respectively, in the spectrum of NG0, indicating hydrogen bonding interactions between bovine skin gelatin and PNIPAAm [39]. This information proved that PNIPAAm and bovine skin gelatin were highly miscible with each other. Additionally, bands related to new chemical bonds were not obtained in the FTIR spectrum; this indicated that there was only physical interpenetration between bovine skin gelatin and PNIPAAm.

Figure 2 shows the SEM image and DLS data of the as-prepared PN1, PN2, PN3, and PN4, from which the morphology and particle size distribution can be evaluated. The particle size of all PNs exhibits a regular photonic packing in the dry state, and the PNs have highly uniform, smooth spherical surfaces. The average particle sizes of PN1, PN2, PN3, and PN4 are ca. 265 ± 5, 385 ± 5, 790 ± 5, and 1080 ± 5 nm, respectively, indicating that the particle size of PN increased stably with the polymerization time. Furthermore, Figure 3 shows the SEM images of the NG0 without PNs. A completely homogenous surface without phase separation in the dry state was observed; this indicated high miscibility between PNIPAAm and bovine skin gelatin (Figure 3a). Bovine skin gelatin generally forms a porous structure after lyophilization because water molecules are removed from hydrated bovine skin gelatin [40]. In the NG0, the strong hydrogen bonding between PNIPAAm and bovine skin gelatin caused high physical interpenetration between them; consequently, bovine skin gelatin occupied the space left by the water molecules (Figure 3a). After blending the PN1 with the NGHH, the embedded PN1 was obviously observed in the NGHH, indicating the high dispersity of PNs in the NGHH matrix (Figure 3b). The PN-embedded NGHHs were immersed in toluene to remove the PNs from them. The cross-linked PNIPAAm network in the NGHH matrix retained the structure after the removal of PNs. Figure 3c shows the SEM images of NG1 after removal of the PNs by toluene post lyophilization at 25 (left) and 37 °C (right), respectively. NG1 exhibited a homogeneous surface without a porous structure at 25 °C, indicating that bovine skin gelatin penetrated into the pores left by PNs to completely fill the pores (Figure 3c left). These results suggest that bovine skin gelatin in the PNIPAAm network still possessed a high degree of freedom to generate chain movement because of the interaction between PNIPAAm and bovine skin gelatin. A slightly porous structure with an average pore size of 93 ± 13 nm appeared at 37 °C, verifying that the extension and shrinkage of bovine skin gelatin sufficiently led to the closing and opening of pores below and above the LCST, respectively (Figure 3c right). For NG2, dispersed pores with ca. 68 ± 9 nm of scale was observed at 25 °C, indicating that the bovine skin gelatin in the PNIPAAm network is not sufficient to fill the larger space left by PNs (Figure 3d left). These pores of NG2 at 25 °C obviously extended from 68 ± 9 to 306 ± 14 nm after increasing the temperature to 37 °C (Figure 3d right). Upon further increasing the particle diameter of PN in the matrix, the pore size of the NG3 increased to 313 ± 18 nm at 25 °C. The pore size less than the PN3 verifies the extension of bovine skin gelatin in the PNIPAAm network (Figure 3e left). Contrarily, the shrinkage of bovine skin gelatin in the PNIPAAm network enlarged the pore size to 786 ± 26 nm at 37 °C (Figure 3e right). An obvious porous structure possessing an average pore size of 1060 ± 74 nm, similar to the particle size of PN4, was observed at 25 °C (Figure 3f left). The average pore size increased from 1060 ± 74 to 1120 ± 25 nm upon switching the temperature from 25 to 37 °C (Figure 3f right).

The results indicate that the bovine skin gelatin penetrates irregularly into the pore left by PN4 (larger than 1 μm) below the LCST. Shrinking of the bovine skin gelatin in the PNIPAAm network led to the uniformity of pores above the LCST. The pore closing and opening behavior of NGHH is appropriate for use as a thermoresponsive gate for smart drug release. The efficiency of pore closing and opening is defended as follows:(5)$$E_{p}=\frac{D_{37{}^{\circ}}{-D}_{25{}^{\circ}}}{D_{37{}^{\circ}}}$$
where *D*_25°_ and *D*_37°_ represent the pore diameter at 25 and 37 °C, respectively. Figure 3g reveals the average pore size of all the NGHH at 25 and 37 °C and the *E*_p_.

For NG1, the *E*_p_ reaches 1, indicating the highest efficiency of pore closing and opening at 25 and 37 °C, respectively. The *E*_p_ of the NGHH decreased with the increasing the space left by PNs. The results suggest that the ca. 265 nm diameter pore left by PNs in the NGHH could be filled completely by bovine skin gelatin in the cross-lined PNIPAAm network below the LCST resulting in pore closing. The closed pore could be opened above the LCST due to the shrinkage of bovine skin gelatin. However, the over 1 μm diameter pore left by PNs in the NGHH did not exhibit high *E*_p_ (0.1). Therefore, the diameter of PNs ranging from 200 to 300 nm may be appropriate to create the pores in the NGHH, leading to a significant thermoresonsive pore behavior. The change in the pore size of the NGHH is predominately determined by the bonding ability of NIPAAm for bovine skin gelatin in the network.

Figure 4 shows the pore opening and closing mechanism below and above the LCST. In the PNIPAAm–gelatin system, the semi-free bovine skin gelatin (green line) in the cross-linked PNIPAAm (orange line) network extends into the pores left by PNs. At a small pore left by PNs in the network, the pores could be completely filled with the bovine skin gelatin and PNIPAAm network below the LCST. Above the LCST, the formation of intermolecular hydrogen bonds among the NIPAAm groups causes the collapse of the bovine skin gelatin and PNIPAAm network in the pores, resulting in the shrinkage of the bovine skin gelatin to the PNIPAAm network and enlargement of the pore size. When the temperature decreases below the LCST, the pores are refilled with the extended bovine skin gelatin because of the swelling of the NIPAAm groups by hydrogen bonding. The reversible formation of the hydrogen bonding network between PNIPAAm and bovine skin gelatin inside the pores leads to thermo-tunable pore sizes, providing satisfactory performances in the further application of smart drug gating.

### 3.2. Surface Performance of the Thermoresponsive NGHH

Hydrophilicity was evaluated by measuring static water contact angle (SWCA) on various NGHH surfaces. Generally, roughness enhances the wettability of a hydrophilic surface, but it reduces the wetability of a hydrophobic surface [41]. Figure 5a shows the SWCAs of these porous NGHHs at 25 and 37 °C, respectively. The presence of the porous structure leads to a significant difference in the SWCA at 25 and 37 °C, respectively, to enhance the thermoresponsive efficiency. The SWCAs of NG0 are 44° ± 4° and 50° ± 4° at 25 and 37 °C, which are attributed to the incorporation of bovine skin gelatin with PNIPAAm, resulting in unobvious thermoresponsive behavior. The SWCA at both 25 and 37 °C increased for all porous NGHH, which is not consistent with previous study [41]. It can be attributed to the cohesion between hydrophilic groups of both PNIPAAm and bovine skin gelatin, which reduces the surface wettability; as a result, the SWCA did not decrease with roughness below the LCST. Figure 5b shows that the reversible behavior was in SWCA for 5 cycles. As we expected, all NGHH samples underwent a stable flipping SWCA from below to above the LCST, and vice versa. The difference of SWCA at 25 and 37 °C enhanced as the roughness increased. For NG4, the SWCA cycled from 68° ± 4° to 111° ± 4° between 25 and 37 °C, indicating the highest thermoresponsive efficiency in SWCA.

The bovine skin gelatin was retained in the cross-linked PNIPAAm network via hydrogen bonding. The stability of the NGHH was investigated by measuring the weight loss post lyophilization before and after immersing these samples in the PBS solution for different incubation times until 20 days (Figure 6a). NG0 exhibited a weight loss of 2.1% ± 0.7% in the first four days. The weight loss of NG0 achieved a plateau in the following sixteen days, indicating a slow degradation rate of up to 6.8% ± 1.1%. The NG0, NG1, and NG2 showed a similar degradation rate on the first day, verifying that bovine skin gelatin was retained in the cross-linked PNIPAAm via strong hydrogen bonding in the NGHH possessing a relatively lower pore size. A relatively higher degradation rate of the NG1 and NG2 than that of NG0 was observed during the following four days. The degradation progressively increased up to the end of the test (20 days) until a slow degradation rate was achieved, which indicated that the pore structure accelerated the degradation. Therefore, the highest largest pore size (NG4) led to the highest degradation (14.8% ± 3.1%) within 20 days. Considering the 7.1% weight loss of NG0 within 20 days, the weight loss of bovine skin gelatin was ca. 7.7% within 20 days.

PNIPAAm is introduced into the bovine skin gelatin to provide the thermoresponsive behavior for application in biological smart drug gating. To evaluate the biocompatibility of NGHHs, the L929 fibroblasts were seeded on the samples for 3, 6, 12, and 48 h of incubation time at 37 °C. Cell adhesion in the early stage (3 and 6 h) is considered an indicator of biocompatibility that precisely predicts the interaction between the cells and NGHH membranes, via which the enhancement of cell proliferation and differentiation can be finally evaluated [42]. Figure 6b shows the relative cell viabilities, determined by the MTT assay, of pure cross-linked gelatin, NG4, NG3, NG2, and NG1 for 3, 6, 12, and 48 h. The results indicated that the cross-linked PNIPAAm network with bovine skin gelatin did not exhibit significant adverse effects on the relative cell (L929 fibroblast) viability at room temperature. The NGHHs showed similar relative cell viability during the initial 3 and 6 h of cell incubation; this indicated that the cross-linked PNIPAAm network with bovine skin gelatin did not significantly reduce the proliferation and differentiation ability of cells. Moreover, NG4 exhibited higher relative cell viability among these NGHHs during the next 12 and 48 h due to its highest pore size. Developing a porous structured surface is a general strategy to improve the affinity between the cells and the substrate. These results suggest that the cells on pore-rich NGHHs exhibit higher viability than those on the pore-less NGHHs. In addition, hemocompatibility of the antibiotic gating was evaluated for the application in blood. The hemocompatibility of NG1 at different temperatures were investigated via hemolytic and anticoagulant assay, respectively. As shown in the inset of Figure 6b, after the immersion of NG1 to the human red blood cell (HRBC) suspension at the concentrations of 10 mg mL^−1^, both 25 and 37 °C did not lead to any obvious hemolytic effect when compared with the negative control (PBS). In contrast, the positive control of water induces a significant hemolysis of HRBCs.

### 3.3. Thermoresponsive Drug Release of Porous NGHHs

Swelling ratio for most of gels generally represents the drug-loading efficiency of biomaterials according to a swelling-controlled mechanism [43]. Swelling ratios of the porous NGHH from 25 to 51 °C for 24 h were determined gravimetrically (Figure 7a). The swelling ratios of these NGHH were not remarkable, which was attributed to the relatively high degree of cross-linking [44]. An approximately linear drop in swelling ratio occurred from 31 to 37 °C, indicating that the LCST of PNIPAAm is affected by the hydrogen bonding interaction with bovine skin gelatin. The swelling ratio of the NGHH without pores (NG0) decreased from 13.5% to 4.5% upon increasing the temperature from 25 to 51 °C. The swelling ratios of the NGHHs decreased with the increase of pore size at 25 °C, indicating that smaller pores in the NGHH facilitate retaining the bovine skin gelatin in the NGHH below the LCST. The swelling ratio of NG1 with less than 100 nm pores reached a maximum below the LCST. With flipping temperature above the LCST, all the swelling ratios—ranging from 4.5% to 7.2%—of the NGHHs were similar, which was attributed to the collapse of the PNIPAAm and bovine skin gelatin network in the pores. Figure 7b shows the loading density (*D*_load_) of the NGHHs for both of AMX and MB at 25 and 37 °C. For NG0, *D*_load_ values are 1.93 and 1.43 mg/g for AMX and MB at 25 °C, respectively, which represent the absolute amount of drugs in the NGHHs at 25 °C. With flipping the temperature to 37 °C, the *D*_load_ did not decrease significantly, which was attributed to the dense structure without pores. The *D*_load_ values of the NGHHs for both AMX and MB at 25 °C increase with the pore size, indicating that the pores substantially enhance the store efficiency of the drugs at 25 °C. In addition, AMX interacted more strongly with the NGHH than MB, resulting in the higher *D*_load_ value at 25 °C. The *D*_load_ values of all the NGHHs reduce abruptly upon flipping the temperature to 37 °C, implying the thermo-triggered drug release behavior. In contrast, the *D*_load_ values of the NGHHs for both AMX and MB decrease with the increasing of pore size at 37 °C. The *D*_load_ values of NG4 for both AMX and MB reduced significantly from 27.32 and 25.26 mg/g to 5.05 and 4.55 mg/g, respectively.

To examine the pore size effect in smart releasing, the R_cr_ values of these NGHHs were plotted vs. temperature from 25 to 45 °C, as shown in Figure 8a. The NG0 did not retain the AMX at 25 °C well, resulting in the high R_cr_ value that reached 100% at 39 °C. The R_cr_ values of the other porous NGHHs remained constant from 25 to 31 °C, indicating the high ability of drug retaining. The linear increase range of drug release with the increasing temperature is defined as a drug release-tunable zone, which could be applied for tuning the drug dosage by temperature. The LCST of pure PNIPAAm is generally in the narrow range of around 32 °C because of the interaction between PNIPAAm and water molecules [45]. The drug release-tunable zone extended with pore size in the NGHH due to the surface tension of AMX. The drug release-tunable zone of NG1 significantly extended with an increase in temperature from 31 to 41 °C. The results of the tunable drug release verify the pore size effect in drug release. Figure 8b shows the kinetics of AMX release from the porous NGHH in real time over four stages: from 25 °C for 120 min, to 37 °C for 180 min, to 25 °C for 180 min, to 25 °C for 240 min. AMX were loaded into the lyophilized porous NGHH to release at 25 and 37 °C repeatedly.

The loaded AMX in the NG0 released completely within 240 min, verifying that the NGHH without pores did not retain AMX efficiently. The R_cr_ value remained below 5% at 25 °C for 120 min, which verified that the porous structure retained the AMX stably. Obvious increases in the R_cr_ values were observed at 37 °C for all NGHHs due to the extension of pores, indicating the thermo-trigged releasing capability. The R_cr_ vaues of these NGHH reduced instantly with cooling the temperature to 25 °C. Finally, the AMX was released at 37 °C for these NGHHs until R_cr_ values reached 100%. As we expected, the R_cr_ value of NG1 reached 100% within 700 min, indicating the capability of long-term drug release. The antibacterial properties of the thermo-triggered AMX release against both *E. coli* and *S. aureus* were evaluated at 25 and 37 °C, respectively. Figure 9a,b shows the BIR, calculated by Equation (4), of all tested NGHHs at 25 °C against *E. coli* and *S. aureus*, respectively, within a period of 12 h. The BIR values of both NG3 and NG4 reached the maximum within the first 6 h, indicating that *E. coli* and *S. aureus* were inhibited under diffused AMX release from NG3 and NG4. BIR values of NG3 and NG4 decreased to 80.6% and 85.4% within next 6 h, respectively, indicating the dosage reducing of the diffused AMX. For NG1 and NG2, the BIR values increased gradually to plateaus, indicating that a smaller pore structure facilitates the prolongation of diffused AMX release. For NG0 without a porous structure, the highest BIR value was ca. 52.4% and 50.7% for *E. coli* and *S. aureus*, respectively, within the first 4 h due to the lower dosage of drug loading. In addition, AMX possesses the higher efficiency to suppress the growth of *E. coli* than that of *S. aureus*. When the temperature was switched from 25 to 37 °C, all the BIR values of these NGHHs enhanced abruptly for both *E. coli* and *S. aureus* in the first 4 h due to the thermo-trigged release of AMX (Figure 9c,d). The BIR values of NG3 and NG4 decreased gradually to ca. 88% within the next 8 h for both *E. coli* and *S. aureus*, which was attributed to the slight decay of thermo-trigged release. However, the decay of thermo-trigged release did not occur obviously for NG1 and NG2. BIR values of NG1 were 88% and 86.2% for *E. coli* and *S. aureus*, respectively, within the first 2 h, and they increased slightly to 90.4% and 89.5%. Although the NG1 did not exhibit the excellent bacteria-inhibiting effect at the initial stages, the nanoporous structure of NG1 prolonged the inhibiting effect against bacteria within 12 h. To further observe the antibacterial activities, NG1 against *S. aureus* for 6 h at 25 and 37 °C were subsequently explored. As we expected, zones of inhibition for NG1 displayed a higher antibacterial activity at 37 °C than that at 25 °C. (Figure 9e). Meanwhile, their average inhibition diameters toward *S. aureus* were calculated, and the diameter of *S. aureus* for NG1 was 0.84 cm at 37 °C, which was obviously larger than that at 25 °C (0.37 cm). These results clearly suggest that the NGHHs show promise as biocompatible carriers for thermo-trigged smart drug release.

## 4. Conclusions

Herein, bovine skin gelatin and PNIPAAm were blended with different size PNs to generate NGHHs via the formation of a cross-linking network. During the removal of PNs from the NGHHs, bovine skin gelatin in the NGHH extends to occupy the pores left by PNs; this causes a decrease in the pore size below the LCST. Above the LCST, bovine skin gelatin shrinks to enlarge the pores of these NGHHs. Upon adjusting the pore size, various dosages of drugs were loaded within the NGHHs below the LCST. The pore closing and opening below and above the LCST caused a significant change in the drug loading within the NGHHs. The tunable porous structure of these NGHHs is substantially dependent on the temperature; thus, they can be used as thermo-tunable valves of drugs. AMX could be retained within the pore structure by hydrogen bonding interaction below the LCST. Upon switching the temperature above the LCST, AMX could be released from the NGHHs. As a result of the high affinity between PNIPAAm and bovine skin gelatin, the NGHH exhibits high stability, biocompatibility, and excellent closing and opening efficiency of thermo-tunable pores, providing a facile approach for the preparation of a smart gate of drug dosage against inflammation from infection or injury inflammation.

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
