# Peer review of "Thermo-Tunable Pores and Antibiotic Gating Properties of Bovine Skin Gelatin Gels Prepared with Poly(n-isopropylacrylamide) Network"

_polymers, 2020, doi:10.3390/polym12092156_

Round 1
Reviewer 1 Report
The manuscript entitled “Thermo-tunable pores and antibiotic gating properties of bovine skin gelatin gels prepared with poly(n-isopropylacrylamide) network” has been reviewed. It is well presented with informative introduction for readers to follow. The authors have designed tunable porous structures of NGHHs which are dependent on temperature - paving the way to design thermo-tunable valves of drugs. In my opinion, this is simple but an amazing idea to design thermos-responsive drug delivery systems. However, the following concern needs to be addressed before it is accepted for publication.
- It seems the supplementary material is somehow not available for the reviewer, or did miss to find it ! In line Nos. 199 to 212, the authors have referred to the figure S1 which should be present in the supplementary document, was not available during review. In addition, these FTIR data should be provided in the main document.
Author Response
Reviewer #1: The manuscript entitled “Thermo-tunable pores and antibiotic gating properties of bovine skin gelatin gels prepared with poly(n-isopropylacrylamide) network” has been reviewed. It is well presented with informative introduction for readers to follow. The authors have designed tunable porous structures of NGHHs which are dependent on temperature - paving the way to design thermo-tunable valves of drugs. In my opinion, this is simple but an amazing idea to design thermos-responsive drug delivery systems. However, the following concern needs to be addressed before it is accepted for publication.
It seems the supplementary material is somehow not available for the reviewer, or did miss to find it ! In line Nos. 199 to 212, the authors have referred to the figure S1 which should be present in the supplementary document, was not available during review. In addition, these FTIR data should be provided in the main document.
â–²Our reply: We have provided the FTIR as the Figure 1 in the main document.
Reviewer 2 Report
The manuscript entitled “Thermo-tunable pores and antibiotic gating properties of bovine skin gelatin gels prepared with poly(n-isopropylacrylamide) network” written by Tsai et al. is about preparing a thermosensitive drug delivery system capable of releasing amoxicillin (AMX) against bacteria. The authors have carried out an extensive characterization study of the system in terms of FT-IR spectroscopy, SEM microscopy, hydrophilicity, degradation rate, and swelling ratio. In addition, a biocompatibility study, drug release, and antibacterial activity were also discussed. Overall, the manuscript is quite interesting, figures are of good quality, and data are convincing. The topic fits in completely with the aims of the journal. However, I find several issues that prevent me to endorse its acceptance at the present stage.
Minor points
a. Pag. 3. Line 117. Scheme 1. Even though the authors have described the experimental protocol in section 2.2, I recommend the authors describe the preparation process step-by-step in the figure as a legend
b. Pag. 4. Line 163. Drug loading and releasing. The authors should include in the experimental section at which pH the release of AMX was carried out.
Major points
a. The authors have shown the biocompatibility of the antibiotic gates from MTT analysis. It would be interesting to study further their biological safety on whole blood carrying out hemolysis studies. This might confirm the total safety of these systems after parenteral injections.
b. To further confirm the biocompatibility of the antibiotic gates, authors should include in the manuscript pictures/images of the cell morphology after treatment with the antibiotic gates.
c. Besides studying the release of AMX through the antibiotic gates and calculating the bacteria inhibition ratio, the authors should include Petri dish pictures including the images of the inhibition zone. This will help readers confirm the potential of these drug delivery systems as antibiotic gates.
d. Antimicrobial activity of AMX alone should be included as a control to compare the effectiveness of the drug delivery systems prepared in this study

Author Response
Reviewer #2: The manuscript entitled “Thermo-tunable pores and antibiotic gating properties of bovine skin gelatin gels prepared with poly(n-isopropylacrylamide) network” written by Tsai et al. is about preparing a thermosensitive drug delivery system capable of releasing amoxicillin (AMX) against bacteria. The authors have carried out an extensive characterization study of the system in terms of FT-IR spectroscopy, SEM microscopy, hydrophilicity, degradation rate, and swelling ratio. In addition, a biocompatibility study, drug release, and antibacterial activity were also discussed. Overall, the manuscript is quite interesting, figures are of good quality, and data are convincing. The topic fits in completely with the aims of the journal. However, I find several issues that prevent me to endorse its acceptance at the present stage.
Minor points
a.Pag. 3. Line 117. Scheme 1. Even though the authors have described the experimental protocol in section 2.2, I recommend the authors describe the preparation process step-by-step in the figure as a legend
â–²Our reply: We have described the simple preparation process step-by-step in the scheme 1 as a legend.
b.Pag. 4. Line 163. Drug loading and releasing. The authors should include in the experimental section at which pH the release of AMX was carried out.
â–²Our reply: The drug loading and releasing were carried out in PBS solution at pH 7.4. We have addressed the pH value in the experiment section.
Major points
a.The authors have shown the biocompatibility of the antibiotic gates from MTT analysis. It would be interesting to study further their biological safety on whole blood carrying out hemolysis studies. This might confirm the total safety of these systems after parenteral injections.
â–²Our reply: We have provided the hemolysis studies in the Figure 6b with discussion to confirm the total safety of these systems after parenteral injections.
b.To further confirm the biocompatibility of the antibiotic gates, authors should include in the manuscript pictures/images of the cell morphology after treatment with the antibiotic gates.
â–²Our reply: Cells generally change their conformation on the porous surface. In addition, cell culture was carried out in PBS solution including several kinds of components. To observe the cell morphology with SEM in a vacuum chamber, the media of cells were replaced to remain the cell morphology until the samples were dried completely. We have observed the cell morphology with SEM; however, it is difficult to identify the cells from other components. Moreover, this work is focused on the thermoresponsive pore of the antibiotic gate for the treatment of skin infections. Cell morphology may not be significantly important in this work. Our future work may develop the cell culture on the porous surface with the antibacterial gate. Cell morphology will be investigated during cell culture in the future work.
c.Besides studying the release of AMX through the antibiotic gates and calculating the bacteria inhibition ratio, the authors should include Petri dish pictures including the images of the inhibition zone. This will help readers confirm the potential of these drug delivery systems as antibiotic gates.
â–²Our reply: We have included the Petri dish pictures including the images of the inhibition zone for NG1 at 25 and 37 degree C, respectively, as Figure 9e to confirm the potential of these drug release systems as antibiotic gates.
d.Antimicrobial activity of AMX alone should be included as a control to compare the effectiveness of the drug delivery systems prepared in this study
â–²Our reply: Amoxicillin is an antibiotic used to treat a number of bacterial infections. BIR value can reach 100% in a 10 % AMX solution for 12 h. Because released doses from the samples in the solution were unknown, an appropriate AMX concentration is difficult to determine as a control for comparison with other samples. Preparation of an antibacterial agent for treatment of skin infections is our motive. This work is focused on drug loading and releasing of the antibacterial agent by thermo-tunable pores. Therefore, the pore-less NG0 is better to regarded as a control for comparison of the effectiveness with other samples.
Reviewer 3 Report
1. The authors need to include the state of the art related to the work in the introduction. 2. What is the novelty and innovation in the study? 3. How is the study different from the ones currently available in the field? 4. How does the study advance the field? 5. Figure 1, where the authors represent the size from the zetasizer is not completely visible and i suggest to improve the size of the figureAuthor Response
Reviewer #3:
- The authors need to include the state of the art related to the work in the introduction.
â–²Our reply: A paragraph "Skin burns mostly generate the disruption of the epidermal barrier, combined with the denaturation of proteins and lipids, significantly prone to infection due to a fertile environment that is rich in bacterial nutrients for microbial growth[1]. The most abundant species is Staphylococcus aureus (6.5–37.6%) to colonize burn wounds among species present in normal skin microflora [2]. Antibacterial agents for the treatment of skin infections, including burns, in hospitals have favored the emergence of smart antibiotic releasing during serious opportunistic infection, therefore essential[3]." has been included to state the art related to the work in the introduction.
- What is the novelty and innovation in the study?
â–²Our reply: The novelty is "We combined UCST behavior of gelatin and LCST behavior of PNIPAAm to develop a thermoresponsive PNIPAAm-gelatin hybrid hydrogels (NGHHs) for smart gating of drug molecules." The innovation is "During removal of the PNs, the gelated gelatin networks with stretched PNIPAAm chains in the NGHH membranes shrunk the pore sizes below both UCST of gelatin and LCST of PNIPAAm; while the pore enlarged by solating gelatin networks with coiled PNIPAAm chains above both UCST of gelatin and LCST of PNIPAAm. Hereby, the efficiency of thermoresponsive pore has been enhanced significantly." The UCST and LCST behaviors firstly combine to develop the smart drug gate with high efficiency. We have described them in the manuscript.
- How is the study different from the ones currently available in the field?
â–²Our reply: Most of smart drug delivery systems have been developed with particles. Some smart antibacterial and wound healing agents have been prepared with complicated synthesis route. We develop a smart antibacterial and wound healing agents with facile synthesis route to enhance the possibility of mass production.
- How does the study advance the field?
â–²Our reply: There are few articles to report a smart gate that can tune the pore size thermally with high efficiency. We firstly develop high efficiency of thermoresponsive pore for a smart gate.
- Figure 1, where the authors represent the size from the zetasizer is not completely visible and i suggest to improve the size of the figure
â–²Our reply: We have enlarged the size of zetasizer in the Figure 1.
Round 2
Reviewer 2 Report
The authors have addressed all my concerns in the revised manuscript. I recommend publishing the manuscript.
Reviewer 3 Report
Authors have addressed my comments raised previously